# Deciphering the Plastome and Molecular Identities of Six Medicinal “Doukou” Species

**DOI:** 10.3390/ijms25169005

**Published:** 2024-08-19

**Authors:** Ying Zhao, Amos Kipkoech, Zhi-Peng Li, Ling Xu, Jun-Bo Yang

**Affiliations:** 1Plant Germplasm and Genomics Center, Germplasm Bank of Wild Species, Kunming Institute of Botany, Chinese Academy of Sciences, Kunming 650201, China; 18468017154@163.com; 2Germplasm Bank of Wild Species, Kunming Institute of Botany, Chinese Academy of Sciences, Kunming 650201, China; amos@mail.kib.ac.cn (A.K.); lizhipeng@mail.ynu.edu.cn (Z.-P.L.); 3State Key Laboratory for Conservation and Utilization of Bio-Resources, Research Center of Perennial Rice Engineering and Technology, School of Agriculture, Yunnan University, Kunming 650201, China; xw963081882@163.com; 4School of Life Sciences, University of Chinese Academy of Sciences, Beijing 100049, China; 5CAS Key Laboratory for Plant Diversity and Biogeography of East Asia, Kunming Institute of Botany, Chinese Academy of Sciences, Kunming 650091, China

**Keywords:** “Doukou”, DNA barcoding, ITS, medicinal plants, plastome, species identification

## Abstract

The genus *Amomum* includes over 111 species, 6 of which are widely utilized as medicinal plants and have already undergone taxonomic revision. Due to their morphological similarities, the presence of counterfeit and substandard products remains a challenge. Accurate plant identification is, therefore, essential to address these issues. This study utilized 11 newly sequenced samples and extensive NCBI data to perform molecular identification of the six medicinal “Doukou” species. The plastomes of these species exhibited a typical quadripartite structure with a conserved gene content. However, independent variation shifts of the SC/IR boundaries existed between and within species. The comprehensive set of genetic sequences, including ITS, ITS1, ITS2, complete plastomes, *mat*K, *rbc*L, *psb*A-*trn*H, and *ycf*1, showed varying discrimination of the six “Doukou” species based on both distance and phylogenetic tree methods. Among these, the ITS, ITS1, and complete plastome sequences demonstrated the highest identification success rate (3/6), followed by *ycf*1 (2/6), and then ITS2, *mat*K, and *psb*A-*trn*H (1/6). In contrast, *rbc*L failed to identify any species. This research established a basis for a reliable molecular identification method for medicinal “Doukou” plants to protect wild plant resources, promote the sustainable use of medicinal plants, and restrict the exploitation of these resources.

## 1. Introduction

Species identification is crucial in the fields of biology and ecology [1] and serves as the basis for ecological research, enabling the understanding of species richness and biodiversity [2]. It informs conservation efforts through the identification of endangered invasive and keystone species [3]. Additionally, it plays a crucial role in predicting and preventing infectious disease outbreaks by identifying potential disease hosts and transmitters among wild animal species [4]. In food production industries, species identification ensures authenticity, quality, and safety, preventing fraud and the circulation of substandard products [5]. In criminal and forensic cases, it aids in identifying the origin of wildlife products [6]. Traditional methods of species identification, relying on morphological characteristics, have limitations in discriminating taxa with limited morphological differences or complex phylogenetic history. DNA barcoding technology has emerged as an effective advancement to overcome these challenges [7].

DNA barcoding is a molecular technique that identifies biological species by examining distinct DNA segments, utilizing variations in short DNA sequences to provide rapid and reliable species identification [5,8,9,10,11,12]. The concept of DNA barcoding was first proposed by Paul Hebert, who suggested using a small, highly conserved genetic sequence called the “ribosomal RNA gene region” to identify species [5]. Initially, DNA barcoding was widely used in animals, where the gene-encoding cytochrome c oxidase I (COI) in mitochondria has a high species differentiation potential, especially in insects, birds, and fish [13,14]. Therefore, the COI gene has become the preferred choice for universal DNA barcoding in animals due to its high level of accuracy in species identification [15]. However, in plant mitochondrial genomes, the COI gene shows a high degree of conservation and is not suitable as a DNA barcode [16]. In addition, complex evolutionary events such as hybridization, polyploidization, and incomplete lineage sorting are more common in plants than in animals, further increasing the difficulty of screening fragments suitable for DNA barcoding [17]. Currently, the internationally recognized universal plant DNA barcodes include four gene regions, including ITS (internal transcribed spacer: internal transcribed spacer 1-5.8S-internal transcribed spacer (2), *mat*K, *rbc*L, and *psb*A-*trn*H [18]. The selection of these gene regions considers the genetic diversity and evolutionary history of the plant kingdom to enhance the effectiveness and usefulness of plant DNA barcodes. However, these fragments have limitations. As an alternative, ultra-barcoding using the complete plastomes for plant species identification has been proposed [19]. Although discerning closely related species using DNA barcoding can pose challenges, this technique is promising in distinguishing morphologically indistinguishable species but genetically distinct [20].

*Amomum* Roxb. is the second-largest genus in the Zingiberaceae Martinov family after *Alpinia* and includes approximately 111 [21] to 150 [22,23] species distributed in tropical Asia and Australia, particularly in Southeast Asia, such as India, Malaysia, and Indonesia [23]. In China, *Amomum* comprises 39 species (29 endemic and 1 introduced) [23], mainly distributed across the Fujian, Guangdong, Guangxi, Guizhou, Yunnan, and Tibet provinces [22]. Among these, six species are listed in the Chinese Pharmacopoeia [24]. These species, originally classified within the genus *Amomum*, have undergone taxonomic revision [25]. These encompass (1) *Lanxangia tsaoko* (Crevost & Lemarié) M. F. Newman & Škorničk (synonym: *A. tsaoko* Crevost et Lemarie), (2) *Wurfbainia compacta* (Sol. ex Maton) Škorničk. & A. D. Poulsen (synonyms: *A*. *compactum* Sol. ex Maton and *Zingiber compactum* (Sol. ex Maton) Stokes), (3) *W. longiligularis* (T. L. Wu) Škorničk. & A. D. Poulsen (synonym: *A. longiligulare* T. L. Wu), (4) *W*. *vera* (Blackw.) Škorničk. & A. D. Poulsen (synonyms: *A*. *krervanh* Pierre ex Gagnep. and *A*. *verum* Blackw.), (5) *W. villosa* (Lour.) Škorničk. & A. D. Poulsen (synonyms: *A. villosum* Lour., *Cardamomum villosum* (Lour.) Kuntze, and *Z. villosum* (Lour.) Stokes), and (6) *W. villosa* var. *xanthioides* (Wall. ex Baker) Škorničk. & A. D. Poulsen (synonyms: *A*. *xanthioides* Wall. ex Baker, *A. villosum* var. *xanthioides* (Wall.ex Bak.) T. L. Wu & S. J. Chen, and *C. xanthioides* Wall. ex Kuntze) [25]. They exhibit a diverse range of traits and applications. For instance, *W. compacta* is a widely used culinary spice and its fruits, leaves, and seeds have a wide range of pharmacological activities in traditional medicine, such as antifungal, antibacterial, antioxidant, gastroprotective, anti-inflammatory, immunomodulatory, anticancer, antiasthmatic, and treatment of acute renal failure [26]. The fruits of *W. vera* have shown antibacterial activity [27]. The active ingredients in *W. longiligularis* and *W. villosa* var. *xanthioides* have antibacterial activity [28,29]. In addition, the powerful antioxidant properties of *W. villosa* var. *xanthioides* have been used in the treatment of non-alcoholic fatty liver disease (NAFLD) and non-alcoholic steatohepatitis (NASH) [30]. *L. tsaoko* has been found to contain antifungal active substances [31] and antioxidant ingredients [32], indicating its potential medicinal properties; recent studies suggested that it can relieve constipation and could be a promising candidate for developing laxatives [33]. The total flavonoids extracted from *W. villosa* have shown potential for developing new drugs to treat gastric cancer [34]. Chemical components found in the seeds of *W. villosa* can enhance cellular antioxidant activity [35]. Additionally, Chen et al. (2018) have confirmed the potential beneficial effects of *W. villosa* in treating inflammatory bowel disease [36]. Li et al. (2016) demonstrated that the fresh stems and leaves of *W. villosa* can be used as high-quality feed for cattle, sheep, and other grass-eating livestock [37]. Despite these several benefits, their morphological similarities make it challenging and confusing between species. Therefore, molecular identification through DNA barcoding is crucial for accurately identifying *Amomum* species.

Recent studies have utilized various universal barcodes, with the ITS sequence being approximately 500–700 bp long While the length of the ITS sequence is relatively conserved, its sequence exhibits significant variability, making it useful for species differentiation [38]. The sequencing and analysis of ITS are rapid and cost-effective, compared to traditional morphological classification methods. Additionally, an extensive repository of ITS is available in public databases, providing researchers with a wealth of reference resources that facilitate convenient species identification and classification. The GenBank database at the National Center for Biotechnology Information (NCBI) hosts an extensive collection of ITS sequences for *Amomum* and its taxonomic synonyms. As of April 11, 2024, it includes 572 sequences representing 159 species. This vast dataset is a valuable resource for this study, providing comprehensive and diverse information. Selvaraj et al. (2012) revealed that ITS and ITS1 are successful DNA barcodes for differentiating *Boerhavia diffusa* Linnaeus from counterfeit medicinal plants [39]. The ITS2 (internal transcribed spacer 2) region has been utilized for the identification of medicinal plants and their closely related species [40] in the Polygonaceae A. L. Jussieu family [41] and the *Dendrobium* Sw. genus [42]. ITS2 is demonstrated to be the most promising universal DNA barcode for the Zingiberaceae family [43]. The complete plastome, *mat*K, and *rbc*L sequences have shown to effectively distinguish *W. compacta*, *W. longiligularis*, and *W. villosa* [44]. Notably, the *mat*K and the *psb*A-*trn*H intergenic spacer exhibited high identification efficiency for *L. tsaoko* and other *Amomum* species [45]. The most efficient barcodes for the molecular identification of *Amomum* are ITS [46,47,48], ITS1 [49,50], and ITS2 [51,52,53]. These findings have demonstrated the promising potential application of DNA barcoding in species identification and classification within *Amomum*. DNA barcoding can facilitate accurate identification and classification of different *Amomum* species, which helps to understand their diversity and evolutionary relationships, and it is an effective tool that guides methods for *Amomum*’s protection, sustainable utilization, and medicinal value research.

In this study, we employed a combination of newly sequenced data and additional data from the NCBI database, including (1) ITS, (2) ITS1, (3) ITS2, (4) complete plastomes, (5) *mat*K, (6) *rbc*L, (7) *psb*A-*trn*H, and (8) *ycf*1 to facilitate the evaluation and precise identification of six medicinal plants within the *Amomum* genus. Using DNA barcoding, we evaluated the value of barcodes in identifying different *Amomum* medicinal species, thereby reducing the potential errors associated with traditional morphological methods. Our findings will enhance the sustainable management and conservation of *Amomum* resources, thereby facilitating industrial growth and quality control. Ultimately, this will lead to substantial scientific and societal advantages.

## 2. Results

### 2.1. Plastome Structural Variation, Sequence Divergences, and Hypervariable Regions

All 41 individuals from the 6 examined “Doukou” species exhibited a quadripartite structure (Figure 1) and showed limited intraspecific variation in plastome size (Appendix A). The complete plastomes ranged in size from 162,678 to 164,332 bp. The lengths of the large single-copy (LSC), small single-copy (SSC), and inverted repeat (IR) regions ranged from 87,632 to 89,067 bp, 14,895 to 15,754 bp, and 29,642 to 29,971 bp, respectively (Appendix A). There was a slight variation in the total GC content, which ranged from 36.0% to 36.4% (Appendix A). However, the GC content was higher in the IR regions (41.0–41.2%) compared to the LSC (33.7–34.1%) and SSC (29.6–30.3%) regions (Appendix A). The “Doukou” plastomes were highly conserved and encode between 121 and 133 genes, including 82 to 87 protein-coding genes, 8 rRNA genes, and 30 to 38 tRNA genes (Appendix A).

We compared the contraction and expansion of IRs regions at four junctions between the two IRs (IRa and IRb) and the two single-copy regions (LSC and SSC) among six “Doukou” species (Figure 2). The LSC/IRb boundary was embedded in the *rpl*22-*rps*19 region (except for *W. compacta* YWB91902-1 and *W. vera* YWB91901-1, which were directly at the *rpl*22 gene); the IRb/SSC and SSC/IRa boundaries were within the *ycf*1 gene; the IRa/LSC boundary was in the *rp*s19-*psb*A region. The boundary shifts exhibited independent variations both between and within species.

The nucleotide diversity (*Pi*) values were calculated with DnaSP v.5 [54] to test divergence level across different regions within the complete plastomes of the six “Doukou” species and their taxonomic synonyms. The average value of nucleotide diversity (*Pi*) was 0.00469. The nucleotide diversity (π) value ranged from 0 to 0.02354 across the plastomes, and the most hypervariable region was *ycf*1 (Figure 3).

### 2.2. Sequence Characteristics

The matrix characteristics of ITS, ITS1, ITS2, complete plastomes, *mat*K, *rbc*L, *psb*A-*trn*H, and *ycf*1 of six medicinal “Doukou” species and their taxonomic synonyms are listed in Table 1. ITS2 had the highest percentage of variable sites, but complete plastomes had the most variable sites. The same was true for singleton sites (Table 1). ITS1 had the highest percentage of parsimony-informative sites (Table 1).

### 2.3. Distance Based Species Discrimination

Analyses of intra- and interspecific Kimura 2-parameter (K2P) distances identified varying barcoding gaps among six medicinal “Doukou” species and their taxonomic synonyms across different datasets. In a barcoding gap analysis, the ITS1 and complete plastome barcodes exhibited the highest discriminatory power, successfully identifying 50% of the species (3/6; Appendix A; Figure 4. The ITS and *ycf*1 barcode were the next most effective, identifying 33% of the species (2/6; Appendix A; Figure 4). The ITS2 and *psb*A-*trn*H barcodes could only identify one species each, accounting for 17% of the species (1/6; Appendix A; Figure 4). The *mat*K and *rbc*L barcodes could not identify any species (Appendix A; Figure 4. In the ABGD analysis, ITS and ITS1 performed best (3/6; 50%; Appendix A), followed by *ycf*1 (2/6; 33%; Appendix A); while the other five performed the worst (1/6; 17%; Appendix A). The number of generated OTUs varied across the ABGD analysis with the different prior intraspecific divergence in the initial and recursive partitions (Appendix A).

### 2.4. Tree Based Species Discrimination

In the ITS dataset, due to the abundance of sequences for *W. villosa* (synonyms: *A. villosum, C. villosum, and Z. villosum*), maximum likelihood (ML), and Bayesian inference (BI) trees were initially constructed for all individuals (Appendix A). Subsequently, three individuals from the *W. villosa* (*A. villosum, C. villosum, and Z. villosum*) branch of the ML tree were chosen to contribute to the construction of subsequent ITS, ITS1, and ITS2 trees. Similarly, in the *mat*K (Appendix A) and *rbc*L (Appendix A) datasets, three individuals were selected from the same branch in the ML tree for the construction of subsequent *mat*K and *rbc*L trees. In both cases, these individuals were chosen from the top, middle, and bottom of the branch to represent the full range of genetic diversity.

The ML and BI topologies derived from seven of the eight datasets for the six species were congruent in showing which species were monophyletic (Figure 5, Figure 6 and Appendix A), except the ITS1 dataset, which differed from the others (Figure 7 and Appendix A). Across all datasets, including ITS, ITS1, ITS2, complete plastome, *mat*K, *rbc*L, *psb*A-*trn*H, and *ycf*1, *L. tsaoko,* and all individuals of its taxonomic synonyms, formed a monophyletic group, demonstrating the successful identification of *L. tsaoko* (Figure 5, Figure 6, Figure 7 and Appendix A). Similarly, *W. compacta*, along with all individuals of its taxonomic synonyms, formed a monophyletic group in the ITS, ITS1, complete plastome, and *ycf*1 datasets (Figure 5, Figure 6, Figure 7 and Appendix A). In the ITS, ITS1, and complete plastome datasets, individuals of *W. vera* and its taxonomic synonyms exhibited a monophyletic group (Figure 5, Figure 6, Figure 7 and Appendix A). Overall, the ITS, ITS1 and complete plastome datasets can successfully identify *L. tsaoko*, *W. compacta,* and *W. vera* (3/6; Figure 5, Figure 6, Figure 7 and Appendix A); *ycf*1 can successfully identify *L. tsaoko* and *W. compacta* (2/6; Appendix A); the ITS2, *mat*K, and *psb*A-*trn*H datasets can successfully identify *L. tsaoko* (1/6; Appendix A); the *rbc*L dataset cannot identify any species (Figure 8 and Appendix A). However, *W. longiligularis*, *W. villosa*, *W. villosa* var. *xanthioides,* and their individuals of taxonomic synonyms did not form monophyly in the four datasets (Figure 5, Figure 6, Figure 7 and Appendix A).

## 3. Discussion

### 3.1. Plastome Characteristics and DNA Barcode Performance

The plastomes of “Doukou” were highly conserved and exhibited a typical quadripartite structure, a characteristic shared with nine species within the subfamily Zingiberoideae. [55], *Zingiber* Boehm. [56], various species of *Curcuma* L. [57], and other photosynthetic angiosperms [58,59,60]. In the six medicinal “Doukou” species, the maximum possible species discrimination was 3/6 because *W. longiligularis*, *W. villosa*, and *W. villosa* var. *xanthioides* were non-monophyletic for ITS, ITS1, plastome, and plastome-standard barcodes (Figure 5, Figure 6, Figure 7 and Appendix A). 

Taxon-specific markers present a feasible alternative that balances the costs of comprehensive super-barcodes, such as whole plastomes, against the limited genetic variability often found in standard barcodes. For the six medicinal “Doukou” species, we identified the most significant mutational hotspots as the *ycf*1 gene with a π value of 0.02354 (Figure 3), similar to other members of the Zingiberaceae family [46,57]. This was consistent with the *ycf*1 (Figure 8 and Appendix A), having a higher identification rate than the *mat*K, *rbc*L, and *psb*A-*trn*H barcodes. The four conventional barcodes (ITS2, *mat*K, *rbc*L, and *psb*A-*trn*H) were each only able to reliably identify a single species at most, so the *ycf*1 gene region could serve as a viable alternative for species identification for the revised *Amomum* species. Given the financial demands of complete plastome sequencing, this gene region can offer a cost-effective and efficient method for population genetic research on *Amomum*. Additionally, this approach aids in the development of a growing database for taxon-specific barcodes.

### 3.2. Performance Comparison of Species Delimitation Methods

Consistent with previous research [46,47,48], species delimitation outcomes vary with the data and methodologies applied. Among the methods evaluated—ABGD, BG, BI, and ML—ML stood out as the most effective species identification, closely followed by BI, as illustrated in Figure 8. Additionally, the topological structures produced by ML and BI are largely similar, suggesting that these methods consistently achieve the highest identification rates (Figure 8). While the identification rates for ABGD and BG differ, ABGD generally outperforms BG (Figure 8), leading to a method ranking of ML > BI > ABGD > BG. Given the demonstrated robustness and efficiency of ML and BI in this study, these methods are recommended as the preferred approaches for species delimitation in DNA barcode-based identification, particularly when employing super-barcodes.

### 3.3. DNA Barcoding in Six Medicinal “Doukou” Plants

Previous studies have indicated that the standard barcodes are not sufficient for the identification of several medicinal plants within the genus *Amomum* [44,48,49,50,51,53,61]. The complete plastomes have demonstrated a strong capability to differentiate species of *Amomum* [44]. The results of this study have further validated these findings. The ability to distinguish species of *Amomum* is enhanced by the length of the complete plastome sequence, which is approximately 160,000 bp long, and its inclusion of many informative sites. However, the sequencing and analysis of the complete plastomes are considerably more expensive and resource-intensive than short fragments such as ITS sequences. ITS sequencing is more cost-effective and demands fewer computational resources for analysis. Despite its relatively short length of approximately 600 bp, the informative sites within the ITS region can accurately distinguish among *L. tsaoko*, *W. compacta*, and *W. vera*, similar to the capabilities of ITS1. ITS2 can only successfully identify *L. tsaoko*. Although ITS2 contains the highest proportion of variable sites, the complete plastomes hold numerous variable sites (Table 1).

Previous studies have shown that the identification rate of ITS/ITS1/ITS2 is higher compared to the plastome fragments [51,62,63]. This may be because the plastome only contains maternal genetic information [63], while ITS/ITS1/ITS2 contain richer biparental genetic information [64]. ITS sequences typically have multiple copies, which may increase ITS variability and improve the accuracy of species identification. Conversely, plastome fragments may only have a single copy, limiting their identification capabilities in some taxa. Notably, some taxa may contain hybrid or show hybridization leading to difficulties in species identification using plastome fragments. In this case, ITS sequences may better reflect the genetic differences between such species, thereby improving the identification rate.

In the eight datasets, some individuals were placed within monophyletic groups (Figure 5, Figure 6, Figure 7 and Appendix A), possibly due to misidentification. The inclusion of some non-target species individuals in the monophyletic branches might be due to errors in species identification, given that the NCBI database has an extensive range of sources. Previously, several studies solely relied on distance for species identification. However, subsequent research has indicated that the barcode gap may be a result of errors in under-sampled taxonomic groups [65]. Therefore, when carrying out species identification, it is important to incorporate extensive analyses. The relationship between the minimum interspecific distance and the maximum intraspecific distance among the six species, along with the consistency between the ABGD grouping results and the tree results, provides strong evidence to support the species identification and classification of these species. 

### 3.4. NCBI Database as a Resource and ITS vs. ITS1 Identification Rate

The NCBI database has provided comprehensive biological and biomedical information [66], offering a vast collection of genetic sequences, gene expression data, protein structures, and scientific literature. Its user-friendly interface and open-access policy promote global scientific collaboration. However, challenges include navigating through the extensive data and ensuring the quality and accuracy of the information due to varying submissions from researchers and institutions. This research was conducted based on a large amount of NCBI data, and reliable results were obtained. The NCBI database provides great convenience for research. 

The ITS region has been proposed as a standard DNA barcode marker in fungi [67] and plants [68]. In our study, the identification rate of ITS1 was higher than that of ITS2, which aligned with the view that ITS1 is a better barcode than ITS2 in eukaryotes [69]. Despite the evaluation of ITS1 and ITS2 as meta-barcode markers for fungi [70], their identification efficacy as DNA barcode markers varies across different taxa. In this study, the individuals used across the ITS, ITS1, and ITS2 datasets are consistent. Therefore, this research serves as a reference, suggesting that the ITS1 dataset might be considered first in practical applications when the experimental individuals are identical.

Although ITS is significantly longer than ITS1, there is no noticeable difference in the difficulty of amplification between the two. Even though the ITS dataset had more variable sites than ITS1, it did not necessarily mean that it surpasses ITS1 in identification rate. Notably, the percentage of variable sites within the ITS1 sequence is higher than that of variable sites within the entire ITS sequence. Many factors can affect the identification rate, including the presence of key variable sites that can significantly distinguish species, the amount of recognizable feature information within the dataset, and the size of the differences in the species being identified. In these aspects, the ITS1 dataset may be superior to ITS and, thus, have a higher identification rate.

The ITS1 sequence is approximately 100–200 bp in length. In most cases, it can be easily acquired through sequencing of polymerase chain reaction (PCR) amplicons. This process is cost-effective and can easily be obtained. Furthermore, abundant ITS sequences of the genus *Amomum* and its taxonomic synonyms can be directly extracted from the NCBI database. Through multiple analyses in this study, it has been mutually verified that ITS1 has the highest identification rate. This suggests that in future identification of these six medicinal “Doukou” plants, ITS1 should be considered first.

## 4. Materials and Methods

### 4.1. Taxon Sampling

Based on the phylogenetic relationships of the genus *Amomum* established by Boer et al. [25], we selected close relatives of six target species for our study. We consulted this study to identify the accepted names and synonyms for these six target species and their closely related species. Our data collection and analysis focused on these target species, their close relatives, and the taxonomic synonyms associated with both groups. We sampled 11 individuals from *Wurfbainia* and *Lanxangia* genera (Table 2), as well as numerous individuals represented by ITS, complete plastome, *mat*K, *rbc*L, *psb*A-*trn*H, and *ycf*1 sequences from both *Wurfbainia* and *Lanxangia* and their taxonomic synonyms available on NCBI (Appendix A). To download the second-generation sequencing data within these groups, we utilized the prefetch tool in SRA Toolkit v.3.1.0, accessible at https://github.com/ncbi/sra-tools, accessed on 4 March 2024, from the NCBI database. The cut-off date for downloading data from NCBI was 11 April 2024. The detailed species information that was sequenced is listed in Table 2, and all the information was uploaded to the NCBI GenBank database. *Alpinia nigra* (Gaertn.) Burtt (MF076960) and *Alpinia galanga* (L.) Willd. (AF478715) were chosen as outgroups for constructing the matrices of ITS, ITS1, and ITS2 sequences. For the complete plastome, *mat*K, *rbc*L, *psb*A-*trn*H, and *ycf*1 matrices, *A. nigra* (MK940826) and *A. galanga* (MK940825) were selected as outgroups. The selection of outgroups was based on Gong et al. [53]. We downloaded 232, 31, 138, 224, 53, and 31 sequences of ITS/ITS1/ITS2, complete plastome, *mat*K, *rbc*L, *psb*A-*trn*H, and *ycf*1, respectively, from NCBI (Appendix A). Numerous sequences of *W. villosa* (synonyms: *A. villosum*, *C. villosum*, and *Z. villosum*) were recovered for the ITS, *mat*K, and *rbc*L datasets. Initially, we constructed phylogenetic trees using all available data and subsequently selected three individuals from the *W. villosa* clade within the tree based on genetic distance for further analysis.

### 4.2. DNA Extraction, Sequencing, Assembly, and Annotation

We extracted total DNA from 0.2 g of the gel-dried leaves and herbarium samples using the modified 4 × CTAB method [71]. The quality of DNA was assessed using 1% agarose gel electrophoresis and a NanoDrop^®^ ND-1000 spectrophotometer. We constructed a DNA library (300–500 bp) using the NEBNext UItra II DNA library prep kit for Illumina and performed two-end sequencing (2 × 150 bp) on the DNBSEQ-T7 high-throughput platform, generating a total amount of data of no less than 3 Gb. The length of single-ended sequencing reads was 150 bp (sequencing strategy PE150). To convert SRA files downloaded from NCBI into FASTQ format, we used fasterq-dump from SRA Toolkit v.3.1.0 (https://github.com/ncbi/sra-tools, accessed on 4 March 2024). Then, we compressed the ‘fastq’ files into ‘fastq.gz’ format suitable for GetOrganelle assembly using the open-source tool pigz v. 2.2.5 (https://zlib.net/pigz/, accessed on 4 March 2024).

The ITS sequence, spanning approximately 600–700 bp, was first assembled utilizing GetOrganelle v.1.7.5.3 [72]. Following assembly, the resultant FASTG file and the reference from *A*. *sericeum* Roxb. (KY438097.1) were aligned using the Map function in Geneious v.9.0.2 [73] to prepare the sequence for annotation. Subsequently, annotation was performed through Geneious v.9.0.2 [73] with the reference to acquire the ITS sequence. ITS1/ITS2 sequences were then extracted based on annotation information using Geneious v.9.0.2 [73]. 

The plastome assembly and annotation methods of sequences were conducted following the protocol described by Li et al. [74]. The clean data obtained from high-throughput sequencing were directly assembled using GetOrganelle v.1.7.5.3 [72], and the complete circular plastid genome was automatically generated. In cases where the circular structure could not be obtained, results were visually inspected using Bandage v.0.8.1 [75]. Subsequently, reliable plastid genome contigs or scaffolds were identified by manually removing non-target contigs from the ‘fastg’ file. The selected sequences were manually edited and spliced to obtain a complete plastid genome. Annotation of the plastid genome was performed using Geneious v.9.0.2 [73], with the published genome of *A. krervanh* (NC_036935.1) as the reference, and then combined with ORF (open reading frame) for correction. The *mat*K, *rbc*L, *psb*A-*trn*H, and *ycf*1 were extracted using Geneious v.9.0.2 [73] based on annotation information.

The ITS, ITS1, ITS2, complete plastome, *mat*K, *rbc*L, *psb*A-*trn*H, and *ycf*1 matrices were constructed by aligning the sequences using the Mafft Multiple Alignment plugins in Geneious v.9.0.2 [73]. All annotated sequences were uploaded to GenBank, and accession numbers were assigned (Table 2).

### 4.3. Data Analysis

#### 4.3.1. Plastome Structural Variation, Divergence, and Mutational Hotspot Analyses

In this study, we conducted a detailed examination of 41 plastomes from six medicinal “Doukou” species and their taxonomic synonyms, focusing on aspects such as genome size, gene content, which includes protein-coding genes, tRNAs, rRNAs, and GC content. We utilized Geneious v.9.0.2 [73] for comparative analyses to investigate the expansion and contraction dynamics of the inverted repeats (IRs) at the four junctions of these plastomes, with visualization facilitated by IRscope [76]. Furthermore, by employing a sliding window analysis using DnaSP v.5 [54], with settings adjusted to a step size of 200 bp and a window length of 600 bp, we successfully pinpointed the top three sequences as the most variable regions. To complement our research findings, we constructed a detailed physical circular map of the plastome using OGDRAW v.1.3.1 [77].

#### 4.3.2. Sequence-Based Analyses

We conducted a distance-based analysis using matrices generated from a subset of target and closely related species individuals selected from all individuals of the *Wurfbainia* and *Lanxangia* genera and their taxonomic synonyms for tree construction according to Boer et al. [25]. Two primary species delimitation approaches were employed: barcoding gaps (BG) [78] and automatic barcode gap discovery (ABGD) [79]. To investigate the existence of barcoding gaps within each dataset (ITS, ITS1, ITS2, complete plastome, *mat*K, *rbc*L, *psb*A-*trn*H, and *ycf*1), we conducted pairwise distance calculations implemented in MEGA-11 [80] using the K2P model. A scatter plot was employed to identify barcoding gaps by visualizing the relationship between the minimum interspecific distance and maximum intraspecific distance for the six species and their taxonomic synonyms. A species is considered accurately identified when the minimum interspecific distance is larger than its maximum intraspecific distance [81]. The ABGD analysis was conducted using an online platform (https://bioinfo.mnhn.fr/abi/public/abgd/, accessed on 4 March 2024), employing three distinct distance models: Jukes–Cantor [JC69], Kimura [K80] TS/TV 2.0, and simple distance. The analysis was configured with the following parameters: Pmin = 0.001, Pmax = 0.1, Steps = 10, X = 1.5, and Nb bins = 20. The best partition was identified as the one most closely aligning with the delimitation of nominal species among the partitions obtained.

#### 4.3.3. Phylogenetic Tree-Based Analyses

We constructed phylogenetic trees based on ML and BI methods from eight datasets: (1) ITS, (2) ITS1, (3) ITS2, (4) complete plastome, (5) *mat*K, (6) *rbc*L, (7) *psb*A-*trn*H, and (8) *ycf*1 sequences. The sequence matrices of each dataset were aligned using MAFFT implemented in Geneious v.9.0.2 [73]. The ML tree was constructed using RAxML v.8.2.11 [82] by the GTRGAMMAI model with 1000 rapid bootstrap replicates. MrBayes v.3.2.7 [83] was utilized for BI analyses runs with 1,000,000 generations, employing the best-fit model specified according to the optimal scheme selected by jModeltest v.2.1.7 [84] using the Akaike information criterion (AIC) criteria. Phylogenetic trees were then visualized by tvBOT v.3.0 [85]. Successful identification was considered when all individuals of the same species and their synonyms cluster into a single clade.

## 5. Conclusions

In this study, we examined the structural variations in plastomes and assessed the effectiveness of both standard and super DNA barcodes in species identification, focusing on intraspecific and interspecific variability within six medicinal “Doukou” species. Molecular identification of these *Amomum* species was achieved through the analysis of wide genetic markers, including ITS, ITS1, ITS2, complete plastome, *mat*K, *rbc*L, *psb*A-*trn*H, and *ycf*1 sequences. Among the markers employed, ITS, ITS1, and complete plastome were highly effective in identifying *L. tsaoko*, *W. compacta*, and *W. vera*. The *ycf*1 barcode proved useful for identifying *L. tsaoko* and *W. compacta*. In contrast, ITS2, *mat*K, and *psb*A-*trn*H were specifically effective only for identifying *L. tsaoko*. Conversely, *rbc*L was ineffective in distinguishing any of the species. In conclusion, the ITS, ITS1, and complete plastomes performed best followed by *ycf*1, and then ITS2, *mat*K, and *psb*A-*trn*H, while *rbc*L performed worst with insufficient sites to discriminate any species. Consequently, considering factors such as cost-efficiency, ITS1 emerges as the most recommended marker for molecular identification within the *Amomum* genus. The methodologies utilized herein for the molecular identification of the six medicinal “Doukou” species form a basis for the conservation of wild plant resources, the rational utilization of medicinal plants, and the prevention of resource misappropriation. This study provides essential molecular tools for the precise identification of species, hence enhancing our understanding of the botanical and pharmacological aspects of “Doukou” medicinal plants.

## Figures and Tables

**Figure 1 ijms-25-09005-f001:**
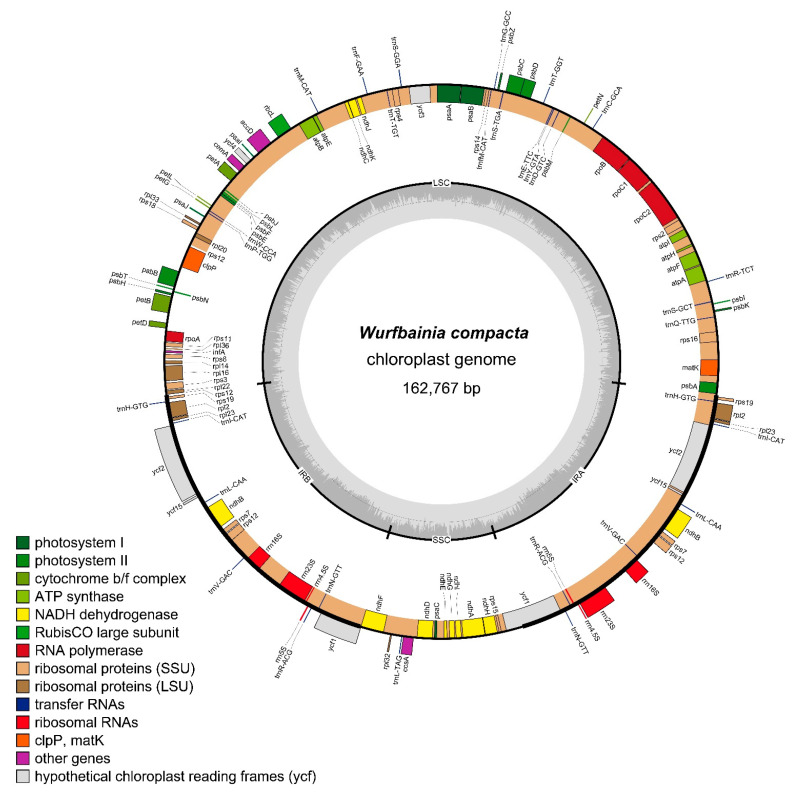
Plastome gene map of *Wurfbainia compacta* YWB91902-2 showing the typical structure organization in “Doukou” plastomes. Genes inside the circle are transcribed clockwise, and those outside are transcribed counterclockwise. Genes in different functional groups are color-coded. The small and large single-copy regions (SSC and LSC) and inverted repeat (IRa and IRb) regions are noted in the inner circle.

**Figure 2 ijms-25-09005-f002:**
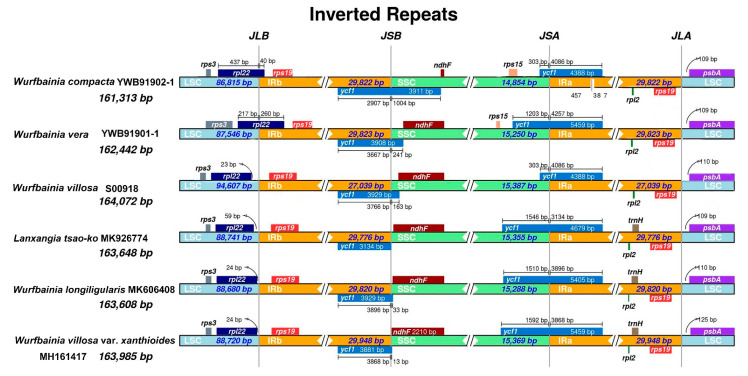
Comparison of the borders of the LSC, SSC, and IR regions among six plastomes of *Amomum*. Abbreviations: JLB—Junction of large single-copy and small single-copy regions; JSB—Junction of small single copy and inverted repeat B; JSA—Junction of small single copy and inverted repeat A; JLA—Junction of large single copy and inverted repeat A.

**Figure 3 ijms-25-09005-f003:**
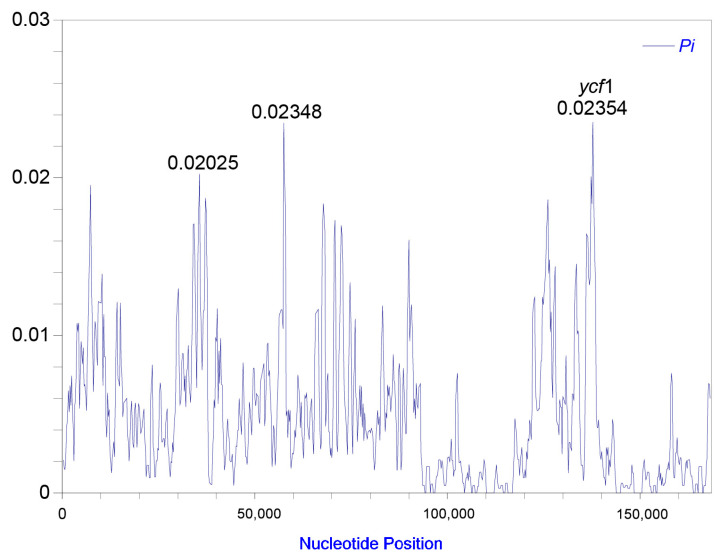
The variable sites in the homologous regions of 41 plastomes of “Doukou” species and their taxonomic synonyms. The *y*-axis represents the nucleotide diversity (*Pi*), and the *x*-axis indicates the nucleotide midpoints.

**Figure 4 ijms-25-09005-f004:**
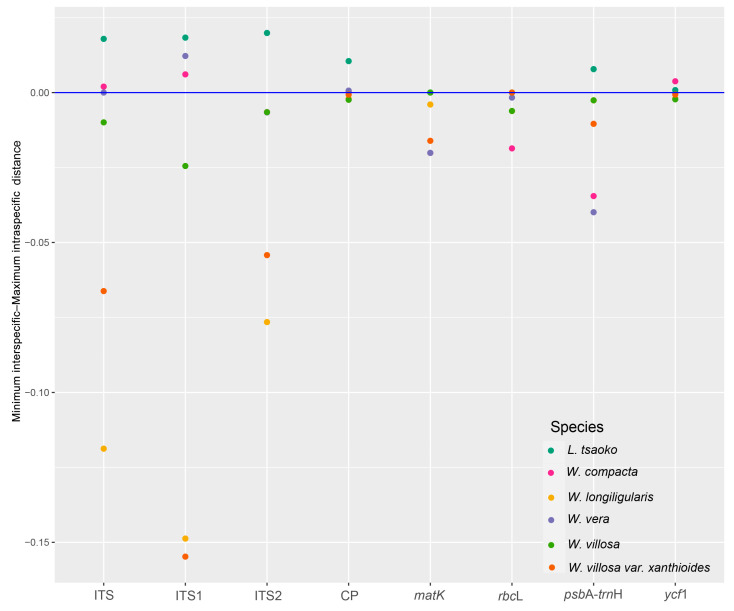
Scatter plot of barcoding gap analysis of the eight datasets across the six medicinal “Doukou” species and their taxonomic synonyms. The *y*-axis represents the genetic divergence, with the plots above the blue line of best fit representing successfully delimited species and those along and below the line representing the overlap. “CP” represents complete plastome.

**Figure 5 ijms-25-09005-f005:**
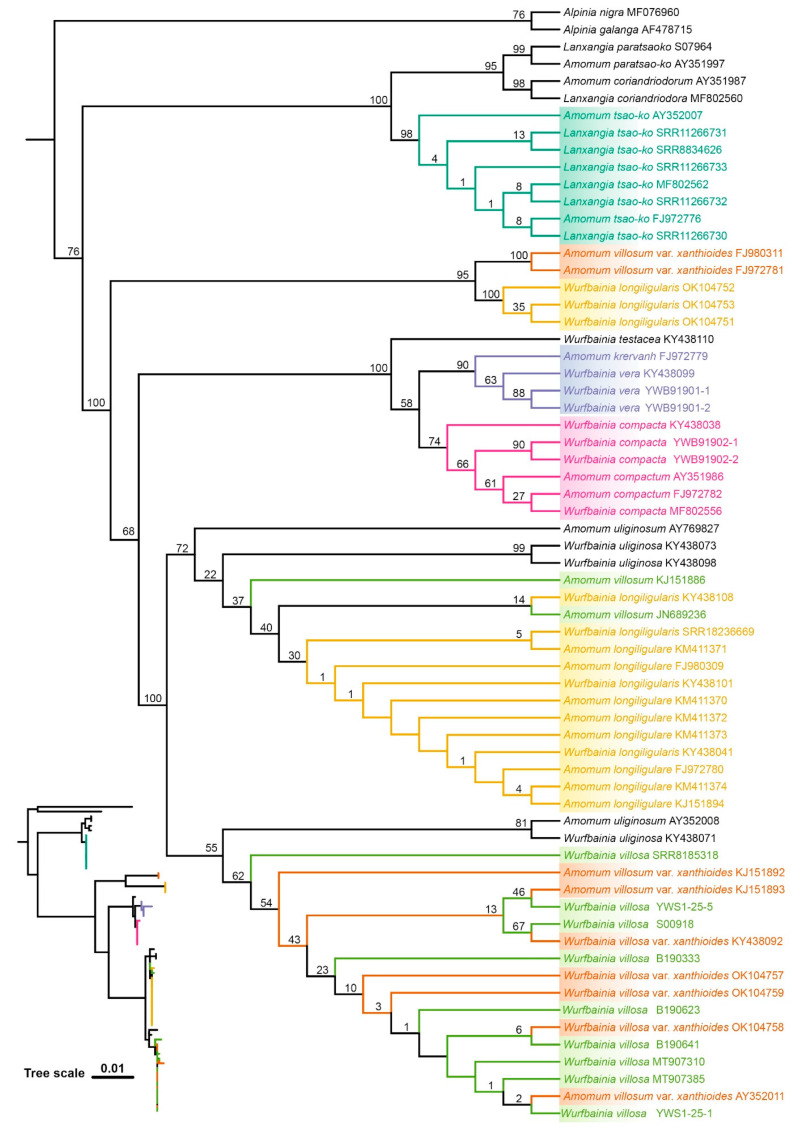
The phylogenetic tree was reconstructed using the maximum likelihood (ML) method with the ITS dataset of six medicinal “Doukou” species and their taxonomic synonyms. The numbers at nodes indicate bootstrap values.

**Figure 6 ijms-25-09005-f006:**
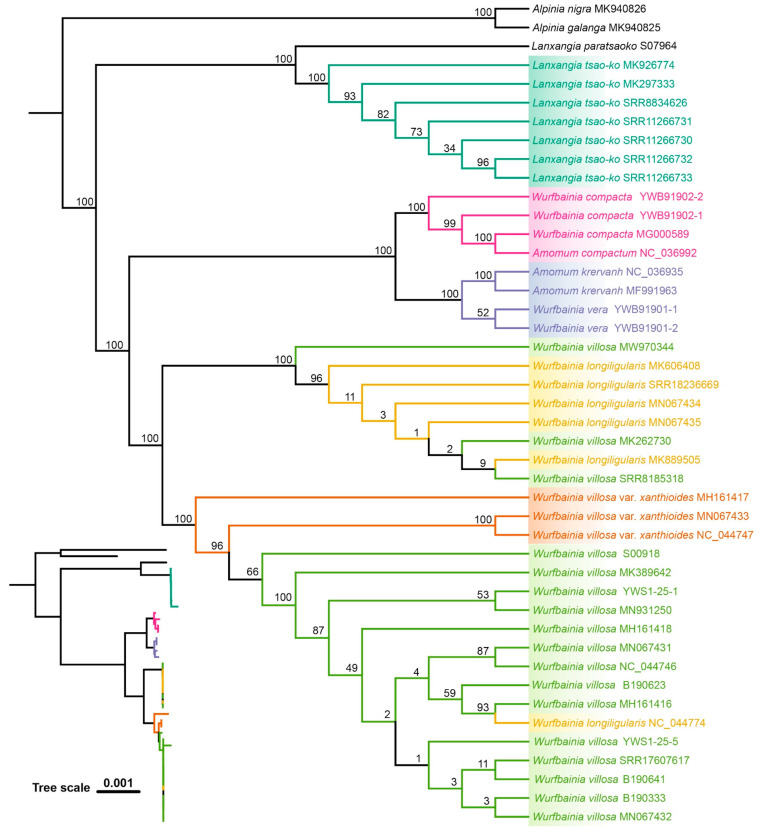
The phylogenetic tree was reconstructed using the maximum likelihood (ML) method with the complete plastome dataset of six medicinal “Doukou” species and their taxonomic synonyms. The numbers at nodes indicate bootstrap values.

**Figure 7 ijms-25-09005-f007:**
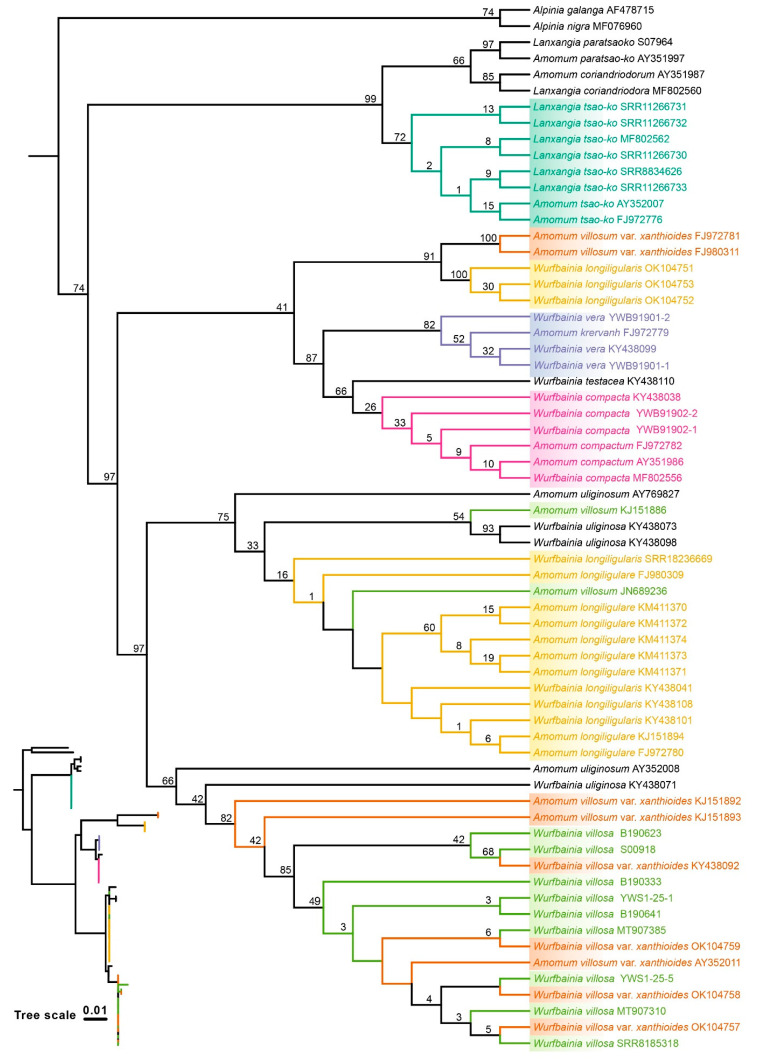
The phylogenetic tree was reconstructed using the maximum likelihood (ML) method with the ITS1 dataset of six medicinal “Doukou” species and their taxonomic synonyms. The numbers at nodes indicate bootstrap values.

**Figure 8 ijms-25-09005-f008:**
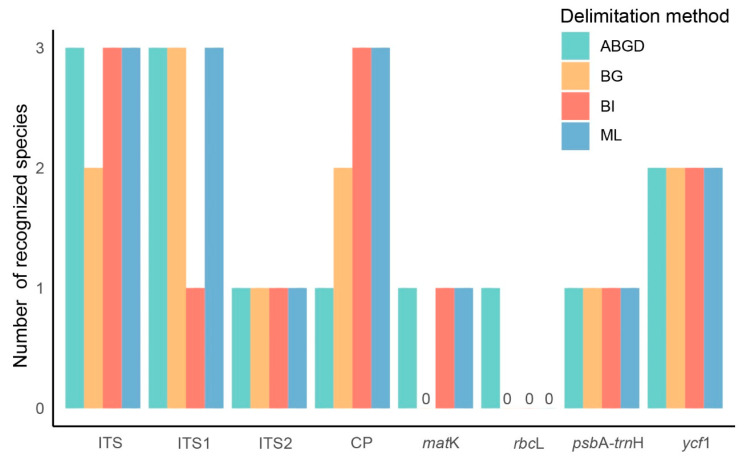
The species discrimination success for candidate barcodes of six medicinal “Doukou” plants across different delimitation methods. The success rate is the number of species successfully delimited to species in the different DNA markers. “CP” represents complete plastome.

**Table 1 ijms-25-09005-t001:** Comparison of characteristics of seven datasets in six medicinal “Doukou” plants.

Dataset	No. of Samples	Aligned Length (bp)	No. of Variable Sites (% Divergence)	No. of Parsimony Informative Sites (% Divergence)	GC Content (%)	No. of Conserved Sites (% Divergence)	No. of Singleton Sites (% Divergence)
ITS	65	609	164 (26.9)	120 (19.7)	56.2	422 (69.3)	42 (6.9)
ITS1	65	194	70 (36.1)	59 (30.4)	56.5	111 (57.2)	11 (5.7)
ITS2	65	222	83 (37.4)	58 (26.1)	60.1	130 (58.6)	23 (10.4)
Complete plastomes	44	168,519	5299 (3.1)	3280 (1.9)	36.1	161,202 (95.7)	1980 (1.2)
*mat*K	82	716	44 (6.1)	28 (3.9)	28.7	672 (93.9)	16 (2.2)
*rbc*L	61	490	12 (2.4)	9 (1.8)	43.2	478 (97.6)	3 (0.6)
*psb*A-*trn*H	66	804	65 (8.1)	35 (4.4)	29.2	690 (85.8)	30 (3.7)
*ycf*1	44	7090	162 (2.3)	91 (1.3)	30.9	6832 (96.4)	67 (0.9)

**Table 2 ijms-25-09005-t002:** Detailed collection information of the newly sequenced *Wurfbainia* and *Lanxangia* species.

Sample Number	Species	Country	Province	Region	Locality	GenBank Accession Numbers for Each DNA Region	
						ITS/ITS1/ITS2	CP/*mat*K/*rbc*L/*psb*A-*trn*H
YWB91902-1	*Wurfbainia compacta*	China	Yunnan	Xishuangbanna Dai Autonomous Prefecture	Mengla County	OR801269	PP826179
YWB91902-2	*Wurfbainia compacta*	China	Yunnan	Xishuangbanna Dai Autonomous Prefecture	Mengla County	OR801270	PP826180
YWB91901-1	*Wurfbainia vera*	China	Yunnan	Xishuangbanna Dai Autonomous Prefecture	Mengla County	OR801267	PP826177
YWB91901-2	*Wurfbainia vera*	China	Yunnan	Xishuangbanna Dai Autonomous Prefecture	Mengla County	OR801268	PP826178
S07964	*Lanxangia paratsaoko*	China	Yunnan	Honghe Hani and Yi Autonomous Prefecture	Yuanyang County	OR801266	PP826176
S00918	*Wurfbainia villosa*	China	Guangxi	Fangchengang City	Shangsi County	OR801265	PP826175
B190333	*Wurfbainia villosa*	China	Yunnan	Kunming City	Xishan District	OR801256	PP826171
B190623	*Wurfbainia villosa*	China	Yunnan	Kunming City	Xishan District	OR801257	PP826172
B190641	*Wurfbainia villosa*	China	Yunnan	Kunming City	Xishan District	OR801258	PP826173
YWS1-25-1	*Wurfbainia villosa*	China	Yunnan	Xishuangbanna Dai Autonomous Prefecture	Mengla County	OR801271	PP826181
YWS1-25-5	*Wurfbainia villosa*	China	Yunnan	Xishuangbanna Dai Autonomous Prefecture	Jinghong City	OR801272	PP853448

Note: “CP” represents complete plastome.

## Data Availability

The datasets presented in this study can be accessed at NCBI GenBank; the list of accessions can be found in Table 2 and Appendix A.

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
