# Peer review of "Deciphering the Plastome and Molecular Identities of Six Medicinal “Doukou” Species"

_ijms, 2024, doi:10.3390/ijms25169005_

Round 1
Reviewer 1 Report
Comments and Suggestions for Authors
The article corresponds to the scope of the journal. It addresses important scientific questions. The manuscript under review has many strengths, but I would like to highlight the weaknesses I have noted.
1. The biggest weakness of the article is the jargon style of the text, which complicates the reading and sometimes leads to factual errors. I will give just a few examples, but there are many similar cases in the article.
a) Lines 237-239: "individuals of A. kravanh, along with all synonymous individuals, exhibited a monophyletic group". This statement is radically at contradicts any theory of taxonomy. Neither individuals, nor populations, nor taxa can be synonymous. Only their names can be synonymous. Authors must be consistent with the principles of taxonomy and nomenclature. Jargon may be used in the laboratory, but it is inappropriate in scientific writings.
(b) Lines 32-33. "holding widespread importance". What do the authors mean by this? What is "widespread importance"? Can there be "rare importance", "casual importance"? I suggest that the authors read the whole text consistently and think about the logical structure of the sentence and the logical relationships between words.
c) c) The first sentence of the abstract is misleading. The genus Amomum does not include six species, it includes many species (at least 111), but six (possibly) are considered medicinal plants. The present sentence can be understood to mean that there are only six species in the genus in total. And what is meant by "have a significant historical background"? What is its meaning?
d) Line 203: "varying barcoding gaps within six Amomum plants across different datasets". What do the authors call plants in this context? The text implies that only six individuals are considered, not six species or other taxa. Is this really the case? I think not.
2. I have a number of complaints about the illustrations. Illustrations in scientific articles are not just decoration, they should provide information. In Figure 1, some entries can only be read by enlarging it on the screen to its maximum size. If such an illustration were printed, the entries would not be readable. Unfortunately, in Figure 2, some of the entries in white letters are not visible and, when scaled up, are unreadable because of the low resolution. In particular, white letter entries on a light background are illegible. The illustrations need to be designed in such a way that the information they contain is legible and understandable.
3. Table 2 provides information on the origin of the samples. Why is the origin of the last samples not given? What do the dashes mean? I strongly recommend that the habitats of the plants sampled are also indicated.
4. I consider the most serious shortcoming of the whole article to be the incomprehensible taxonomy. If the authors state that they are studying the genus Amomum, why are they also covering plants of the genus Wurfbainia. Of the six species studied, only one belongs to the genus Amomum and all the others are members of the genus Wurfbainia. Moreover, Amomum krervanh Pierre ex Gagnep., nom superfl. (= Wurfbainia vera (Blackw.) Škorničk. & A.D.Poulsen) is an illegitimate name. I believe that in such studies dealing directly with taxonomic issues, the principles of taxonomy and nomenclature must be followed precisely. And there is absolutely no justification for the use of the illegitimate species name (Amomum krervanh Pierre ex Gagnep., nom superfl.).
In my opinion, as long as the manuscript does not resolve taxonomy issues or does not explicitly justify the authors' use of a taxonomy that differs from the accepted taxonomy (see Taxon, 67. 2018), it is not possible to consider the possibility of accepting a publication.
Comments on the Quality of English LanguageIn terms of taxonomy and nomenclature, the manuscript is very disorganised and incoherent.
Author Response
Thank you for your valuable feedback on our research. We have responded to your comments and made the corresponding revisions. The detailed responses are included in the uploaded PDF document. Please review it at your convenience.

Reviewer 2 Report
Comments and Suggestions for Authors
In this study, chloroplast-based markers plus whole plastome data were tested for their ability to discriminate species within the genus Amomum. In addition to different markers, different analysis methods were compared, with Maximum Likelihood determined to be the best.
The ability to identify plant species with markers is important for plant research and conservation generally. However, there are some inconsistencies between what is shown in figures and what the text describes. This makes it very difficult to interpret the results.
Specific examples:
Lines 178-180: “Specifically, the LSC/IRb boundary is embedded in the 177 rpl22-rps19 region (except for A. compactum YWB91902-1 and A. kravanh YWB91901-1, 178 which are directly at the rpl22 gene); the IRb/SSC and SSC/IRa boundary is within the ycf1 179 gene; the IRa/LSC boundary is in the rps19–psbA region.” Figure 2 does not clearly show these differences. A. kravanh and A. compactum are not the same as each other. In the same figure, abbreviations JLB, JSB, JSA and JLA are not explained in the figure caption.
Figure 3: While the text points out ycf1 as the point of highest variability, there is another peak at ~58000 that appears to have a similar height, but this is not noted. Regarding the ycf1 gene itself, the discussion notes that it could be used when the other tested markers are inadequate, but why was it not used in this study?
Comments on the Quality of English LanguageModerate English language editing is needed.
Author Response

(The authors gave the same response as above.)

Round 2
Reviewer 1 Report
Comments and Suggestions for Authors
Some of the comments made in the previous review have been appropriately corrected, but the most important comment on taxonomy has not been addressed by the authors. The response to the comment is unsubstantiated, or rather it is an attempt by the authors to manipulate data.
For the sake of clarity, I will quote the abstract of the paper they refer to: "Three genera, Conamomum, Meistera and Wurfbainia, are resurrected, and three new genera Epiamomum, Lanxangia and Sundamomum are described, together with a key to the genera and a nomenclatural synopsis placing 384 specific names (incl. all synonyms) into the new generic framework. Of these 129 represent new combinations and 3 are replacement names." (https://doi.org/10.12705/671.2).
In the same publication, the nomenclatural combinations of most of the taxa studied by the authors of this paper were validated:
Wurfbainia compacta (Sol. ex Maton) Škorničk. & A.D.Poulsen in Taxon 67: 29 (2018).
Wurfbainia longiligularis (T.L.Wu) Škorničk. & A.D.Poulsen in Taxon 67: 30 (2018).
Wurfbainia vera (Blackw.) Škorničk. & A.D.Poulsen in Taxon 67: 30 (2018).
Wurfbainia villosa (Lour.) Škorničk. & A.D.Poulsen in Taxon 67: 30 (2018).
Wurfbainia villosa var. xanthioides (Wall. ex Baker) Škorničk. & A.D.Poulsen in Taxon 67: 30 (2018).
The only assumption that can be made is that the authors are unfamiliar with the taxonomy of the species under consideration, or that they are deliberately manipulating the data.
Comments on the Quality of English Language
The text is written in working slang and has not been corrected, despite the fact that it was pointed out in a previous review as a major shortcoming. Everyone in the laboratory may use jargon, but publications must be written in such a way that the language is logically correct.
Author Response
Thank you for your valuable feedback! We have provided a detailed response to your comments in the attachment. Please review it at your convenience. If you have any further questions, feel free to contact us anytime!

Reviewer 2 Report
Comments and Suggestions for Authors
The researchers have adequately addressed my comments from the previous review. The results are now clearer.
Author Response
Thank you for your valuable suggestions and approval of our article!

Round 3
Reviewer 1 Report
Comments and Suggestions for Authors
The paper has been improved compared to an earlier version, but the most important shortcoming has not been removed. On the contrary, the manuscript has been affected by critical factual errors due to a misunderstanding of the principles of taxonomy and nomenclature.
There is no such thing as 'taxa synonyms'. A taxon is defined as 'taxon (taxa). A taxonomic group at any rank (Art. 1.1)' [here and hereafter reference to the Code of Nomenclature; https://www.iapt-taxon.org/nomen/pages/main/glossary.html] . There may be a 'taxonomic synonym' ('heterotypic synonym (taxonomic synonym). A name based on a type different from that of another name referring to the same taxon (Art. 14.4)'. Only the names of taxa are synonyms, not the taxa themselves. I used to think that the authors were using working slang, but now I have the impression that this impression is due to the authors' lack of subject-specific knowledge of taxonomy and nomenclature.
If you accept the taxonomic system, you must follow it, and if you reject it, the reasons for that rejection must be explained.
You cannot justify your acceptance of a more convenient taxonomic concept instead of the correct and generally accepted one just to make it easier to write a paper. It is the duty of the researcher to interpret and critically evaluate all the information gathered by previous researchers and to provide new knowledge.
I really cannot understand, and the authors do not explain in their reply, why they are rejecting a taxonomic concept that they themselves have cited time and again? And why the authors of the article do not want to accept the genus Wurfbainia and analyse their data in accordance with current taxonomy. If there is a lack of taxonomic knowledge, the authors can always find colleagues who know taxonomy and can help them solve the problems.
Comments on the Quality of English Language
Terminology and expressions with terms (especially taxa, synonyms, synonymy) must be significantly edited.
Author Response
Thank you for your valuable feedback! We have provided a detailed response to your comments in the attachment for your review. Please feel free to contact us if you have any further questions or concerns!

Round 4
Reviewer 1 Report
Comments and Suggestions for Authors
I have no further comments after successive rounds of peer review. Despite the fact that, in my opinion, the title of the article does not mean anything to most non-Chinese readers right now, the authors are free to leave it as it is. In the future, I would wish the authors to analyse the fundamental taxonomic publications before proceeding with the study.
Comments on the Quality of English LanguageMinor revisions of English required.